# Improved Post-Thaw Quality of Canine Semen after Treatment with Exosomes from Conditioned Medium of Adipose-Derived Mesenchymal Stem Cells

**DOI:** 10.3390/ani9110865

**Published:** 2019-10-25

**Authors:** Ahmad Yar Qamar, Xun Fang, Min Jung Kim, Jongki Cho

**Affiliations:** 1College of Veterinary Medicine, Chungnam National University, Daejeon 34134, Korea; drahmadqamar@gmail.com (A.Y.Q.); fx2442@gmail.com (X.F.); 2Department of Theriogenology and Biotechnology, College of Veterinary Medicine, Seoul National University, Seoul 08826, Korea

**Keywords:** dog sperm, cryopreservation, exosomes, post-thaw quality

## Abstract

**Simple Summary:**

In this study, we have tried to exploit the potential role of exosomes-derived from canine adipose-derived mesenchymal stem cells in the protection and repair of damaged sperm during cryopreservation. The treatment of sperm with an optimal level of exosomes resulted in improved post-thaw semen quality. This improvement was due to the exosomal proteins that protect the sperm against oxidative damage and induce the repair of membranes and chromatin. The effect of exosomal proteins was also confirmed by higher expression of genes related to the repair of membranes and chromatin accompanied by the lower expression of a gene associated with reactive oxygen species production.

**Abstract:**

Freezing decreases sperm quality, ultimately affecting fertilizing ability. The repair of freeze-damaged sperm is considered crucial for improving post-thaw viability and fertility. We investigated the effects of exosomes derived from canine adipose-derived mesenchymal stem cells on dog sperm structure and function during cryopreservation. The pooled ejaculate was diluted with buffer, without (Control), or with exosomal proteins (25, 50, or 100 µg/mL). Using fresh semen, the determined optimal exosomal protein concentration was 50 µg/mL (Group 2) which was used in further experiments. Post-thaw sperm treated with exosomes were superior to control (*p < 0.05*) in terms of motility (56.8 ± 0.3% vs. 47.2 ± 0.3%), live sperm percentage (55.9 ± 0.4% vs. 45.4 ± 0.4%), membrane integrity (55.6 ± 0.5% vs. 47.8 ± 0.3%), and acrosome integrity (60.4 ± 1.1% vs. 48.6 ± 0.4%). Moreover, expression of genes related to the repair of the plasma membrane (*ANX 1*, *FN 1*, and *DYSF*), and chromatin material (*H3*, and *HMGB 1*) was statistically higher in exosome-treated sperm than control, but the expression of the mitochondrial reactive oxygen species modulator 1 gene was significantly higher in control. Therefore, exosomal treatment may improve the quality of post-thaw dog semen through initiating damaged sperm repair and decreasing reactive oxygen species production.

## 1. Introduction

The freezing of mammalian sperm is considered imperative for the preservation of genetic material from individuals with superior breeding value or threatened with extinction from some disease or natural disaster. In addition, successfully frozen sperm can be used to manage infertility issues, as in patients affected by the spermicidal effects of radiation or chemotherapy [1]. In dogs, the use of cryopreserved semen reduces the problems concomitant with natural breeding, animal transportation [2,3], and international trade [4]. However, the freezing process can exert certain detrimental changes in the morphology of sperm, resulting from thermal, mechanical, chemical, osmotic, and oxidative damage [5,6]. These changes cause lower post-thaw sperm motility, decrease the integrity of the plasma and acrosomal membrane [7], and damage DNA, ultimately decreasing the fertilizing capability of sperm [8]. The factors responsible for reduced fertility of post-thaw sperm include ice formation, high osmotic pressure [9], reactive oxygen species (ROS) generation [6,10,11], and apoptotic pathway activation [12]. 

Freezing greatly preserves the morphology of male gametes by minimizing the metabolic stresses, ultimately improving sperm viability and fertility [13,14]. Most approaches to improve the outcomes of sperm freezing focused on sperm protection rather than the repair and recovery of damaged sperm. While the previously adopted approaches have acceptable outcomes, the fertility of damaged sperm could be improved by the initiation of repair mechanisms. 

Accumulating evidence from extensive investigations indicate that mesenchymal stem cells (MSCs) have therapeutic potential for treating a variety of injuries, diseases, and syndromes. MSCs are thought to play a vital role in regenerative processes through the secretion of extracellular vesicles (EVs) actively involved in cellular communication [15,16]. EVs are membrane-bound structures, including microvesicles, exosomes, and apoptotic bodies. Several reports indicated their presence in almost all the biological fluids, including ascites, blood, cerebrospinal fluid, and semen [17]. Exosomes contain functional molecules including microRNA, mRNA, DNA, lipids, and proteins [18]. Upon attachment to recipient cells, they are internalized, resulting in a wide variety of epigenetic and phenotypic modifications of the recipient cell. These changes affect recipient cell viability, adhesiveness, resistance to environmental factors, and regenerative capacity [1]. Growth factors secreted *via* exosomes, including platelet-derived growth factor (PDGF), vascular endothelial growth factor (VEGF), basic fibroblast growth factor (bFGF), transforming growth factor beta-1 (TGF-β1), keratinocytes growth factor (KGF), and mitogens are reported to accelerate wound healing through paracrine signaling [19]. In addition, EVs found in the male reproductive fluids (prostasomes) of multiple mammalian species are involved in protecting sperm and activating motility, antibacterial, and antioxidant capacities [20]. 

Given the regenerative nature of functionally active molecules secreted by MSCs, we hypothesized that exosomes derived from canine adipose-derived mesenchymal stem cells (Ad-MSCs) may stimulate the repair of sperm damaged during the freezing process. This study aims to determine whether treatment with exosomes increases the post-thaw fertility and viability of dog sperm. 

## 2. Materials and Methods

All chemicals used in this study were purchased from Sigma-Aldrich (St. Louis, MO, USA) unless otherwise stated. 

### 2.1. Animals Used and Semen Preparation

Semen was collected from four sexually mature, healthy male beagles, aged 2–3 years, weighing 7–11 kg (one ejaculate from each male per replicate). Each individual was housed in a separate indoor cage, with animal care facilities and procedures set up in accordance with the standards established by the Committee for Accreditation of Laboratory Animals at Seoul National University. All experimental procedures were carried out in compliance with the Guide for the Care and Use of Laboratory Animals at Seoul National University (approval no. SNU-180731-2). Sperm-rich fractions were collected twice a week. Semen ejaculates with a sperm count ≥ 100 × 10^6^/mL, ≥70% motile sperm, and ≥80% with normal morphology were used.

### 2.2. Canine Ad-MSCs Culture and Conditioned Medium Preparation

Canine Ad-MSCs and AMSC medium (the canine Ad-MSCs culture medium) were provided by Naturecell Co., Ltd. (Seoul, Korea). Frozen canine Ad-MSCs were thawed in a water bath at 37 °C, washed, and re-suspended in AMSC media. The Ad-MSCs then were cultured in a tissue culture dish and incubated in a humidified environment under 5% CO_2_ at 37 °C until 80–90% confluence. Canine Ad-MSCs derived conditioned media (MSC-CM) was prepared from cultured Ad-MSCs of passage 3 and 4 stage upon reaching 80–90% confluence. Supernatants from Ad-MSCs cultured in DMEM medium were collected after 48 h of serum starvation [21].

### 2.3. Isolation of Exosomes

Exosomes were isolated from MSC-CM using Invitrogen Total Exosome Isolation (Thermo Fischer Scientific, Waltham, MA, USA) (Figure 1). The MSC-CM was centrifuged at 2000× *g* for 30 min to remove the cellular debris, and the supernatant was recovered for further use. The required quantity of MSC-CM was transferred to a new tube, and Total Exosome Isolation reagent was added (0.5 mL per mL of MSC-CM), mixed by vortexing to obtain a homogenous solution, and incubated overnight at 2–8 °C. After incubation, the mixture was centrifuged at 10,000× *g* for 1 h at 2–8 °C. The supernatant was discarded, and the exosomes pellet was used immediately.

### 2.4. Transmission Electron Microscopy

Exosomes were identified using transmission electron microscopy [22]. After isolation of exosomes from the conditioned medium, the exosomes pellet was resuspended in nuclease-free water. The fresh exosomes suspension (5-µL) was placed on parafilm applied to a copper 200-mesh Formvar-coated carbon stabilized grids and allowed to absorb for 4–5 min. The gird was wiped off with filter paper, and a 5-µL drop of 1% aqueous uranyl acetate was placed on the grid. After 5 s the drop was wiped off the grid using filter paper and rinsed the gird using deionized water (3 times for 10 s each). Finally, the gird was wiped off with filter paper and dried for 5 min. Transmision electron microscopic analysis was performed at 110 KV using a LIBRA 120 transmission electron microscope (Carl Zeiss, Oberkochen, Germany). 

### 2.5. Determination of Optimal Exosomal Protein Concentration

The pooled ejaculate was washed by centrifugation at 100× *g* for 1 min at room temperature and the supernatant was recovered. The same volume of buffer 1 (24 g/L tris (hydroxymethyl) aminomethane, 14 g/L citric acid, 8 g/L of fructose, and 0.15 g/L kanamycin sulfate in distilled water (pH 6.6, 290 mOsmol)) was added to the semen and centrifuged for 5 min at 700× *g*. The sperm pellet was re-suspended in buffer 1 to achieve 200 × 10^6^ sperm/mL. A sperm count of 100 × 10^6^ /mL was attained with a suitable volume of buffer 2 (6% (*v*/*v*) glycerol, 40% (*v*/*v*) egg yolk, and 54% (*v*/*v*) buffer 1), either without exosomal proteins (control), or with 25, 50, or 100 µg/mL of exosomal protein. Buffer was added in a multi-step loading protocol, adding 14%, 19%, 27%, and 40% of the total calculated volume at intervals of 30 s. 

The effects of exosomal proteins on sperm quality was evaluated by analyzing the semen samples from four independent replicates processed on different days. After incubating the diluted semen from each group at 4 °C for 8–10 h, the motility and kinematic parameters including linearity (%), straightness (%), and amplitude of lateral head displacement (ALH, µm) of the sperm were evaluated. A semen droplet (10-µL) was placed onto a clean pre-warmed slide and mounted with a coverslip. Sperms were examined in five different fields, and the kinematic parameters of at-least 200 motile sperm cells were tracked by using a sperm analysis imaging system (FSA2011 premium edition version 2011; Medical Supply, Gangwon, Korea).

In addition, eosin-nigrosin staining was employed to investigate the viability of sperm from each group, as previously explained. Briefly, a drop of semen (5–10 µL) was mixed in an equal volume of stain on a pre-warmed glass slide. A thin smear of the semen-stain mixture was placed on a new slide and air-dried. We examined 200 sperm per slide, determining whether the membrane was an intact (white staining, Figure 2a) or non-intact membrane (pink staining, Figure 2b).

### 2.6. Freezing and Thawing of Sperm

The second experiment was performed to evaluate the effects of exosomal proteins on the freezing of sperm. Following the above-mentioned protocols for washing and dilution, 0.5 mL straws (Minitube, Tiefenbach, Germany) were filled with diluted semen. Sealed straws were kept at 4 °C for 1 h equilibration, followed by freezing by placing the straws 2 cm above liquid nitrogen (LN_2_) horizontally for 15 min. For storage, the straws were plunged into LN_2_. One week after cryopreservation, the semen was thawed in a water bath at 37 °C for 30 s, then diluted with buffer 1 (1:5, semen: buffer 1) stepwise as described previously. Sperm viability, motility, and kinematic parameters were determined as described in ‘Determination of optimal exosomal protein concentration’. 

### 2.7. Assessment of Sperm Plasma Membrane Integrity

Sperm plasma membrane integrity was analyzed by hypo-osmotic swelling (HOS) assay [23]. HOS solution was composed of 0.735 g sodium citrate and 1.351 g fructose dissolved in 100 mL distilled H_2_O (osmotic pressure ∼190 mOsm/kg). A 50-µL aliquot of semen from each group was mixed with 500-µL of HOS solution and incubated for 30 min at 37 °C. A 5-µL drop of the mixture was placed on a glass slide, and 200 sperm were analyzed for their ability to swell using a phase-contrast microscope (Eclipse Ts 2, Nikon, Walt Whitman Road Melville, NY 11747-3064, USA). The swelling was indicated by the coiled tail, and such sperm were considered to possess an intact plasma membrane.

### 2.8. Assessment of Acrosome Membrane Integrity

Fluorescein isothiocyanate-conjugated peanut agglutinin (FITC-PNA) was used to assess the integrity of the sperm acrosomal membrane [24]. A post-thaw 30-µL drop of semen was smeared onto a clean glass slide. Air-dried smears were fixed with absolute methanol for 10 min at 20–22 °C and allowed to dry. Staining was performed by spreading 30-µL of FITC-PNA solution (100 µg/mL in PBS) onto each smear. Slides then were incubated in the dark under moist conditions for 30 min at 37 °C, rinsed with PBS, air-dried, and mounted with glycerol. Sperm acrosome status was examined using an epifluorescence microscope (Eclipse Ts 2, Nikon). At least 200 sperm per slide were examined and classified according to the presence (strong green fluorescence, Figure 3) or absence (no fluorescence, Figure 3) of an intact acrosome.

### 2.9. Mucus Penetration Test

Surrogate mucus (modified synthetic oviduct fluid) was loaded into flat capillary tubes (80 ± 0.5 mm long, 1.25 ± 0.05 mm wide) that were sealed at one end. The filled capillary tubes were kept in the vertical position for 15 min to remove bubbles and check the tightness of the seal. Then, the capillary tube was inserted into an Eppendorf tube containing 100-µL of semen suspension and laid horizontally for 2 h at room temperature. Sperm that reached 1 cm and 3 cm marker points in the capillary tube were counted. 

### 2.10. Protamine Deficiency Test

Chromomycin A_3_ (CMA_3_) is a fluorochrome that competes with protamine for binding to the minor groove of DNA. Its fluorescence is used to detect protamine deficiency in loosely packed chromatin and correlates with the extent of nicked DNA [25]. CMA_3_ staining was performed to assess the packing quality of sperm chromatin material [26]. Samples of post-thaw washed canine sperm were smeared onto clean glass slides and fixed in methanol/glacial acetic acid (3:1) for 5 min at 4 °C temperature. Both the control and treatment group (50 µg/mL) slides were treated for 20 min with 100-µL of CMA_3_ solution (0.25 mg/mL of McI1vaine buffer supplemented with 10 mM MgCl_2_, pH 7.0). Smears were rinsed in McI1vaine buffer mounted with buffered glycerol and analyzed using an epifluorescence microscope (Eclipse Ts 2, Nikon) with an appropriate filter. CMA_3_-positive sperm with abnormal chromatin packing was indicated by their bright green fluorescent head, as compared to the dull green fluorescent head of CMA_3_-negative sperm (Figure 4).

### 2.11. Relative Quantitative polymerase chain reaction Analysis

Relative quantitative polymerase chain reaction (qPCR) was used to analyze the mRNA expression of plasma membrane repair-related genes (annexin 1 [*ANX1*], dysferlin [*DYSF*], and fibronectin 1, [*FN1*], chromatin material repair-related genes (histone H3 [*H3*], high mobility group protein B [*HMGB*], and a mitochondrial ROS modulator 1 [*ROMO1*]). Briefly, total RNA was extracted from post-thawed sperm from treated and control samples. Real-time qPCR (RT-qPCR) was used to assess transcript abundance (primers listed in Table 1). RNA was extracted using Trizol reagent (Invitrogen, Carlsbad, California, U.S.A), followed by complementary DNA synthesis using the Compact cDNA synthesis kit (SJ BIOSCIENCE, Daejeon, Korea) following the manufacturer’s instructions. Sybr Green Q-PCR Master mix (SJ BIOSCIENCE, Daejeon, Korea) was used to analyze expression levels of RT-qPCR transcripts and the expression of each target gene was quantified relative to that of the internal gene β-actin (*ACTB*) using the equation, R = 2^−[ΔCt sample − ΔCt control]^.

### 2.12. Experimental Design

Experiment 1 aimed to determine the optimal concentration of exosomal proteins required for the preservation of motility and viability of canine semen. Experiment 2 investigated the effects of supplementation of Buffer 2 with the optimal concentration of exosomal proteins on the quality (motility, kinematic parameters, viability, plasma membrane integrity, and acrosome integrity) of post-thaw sperm. Experiment 3 aimed to determine the capability of post-thaw sperm to penetrate mucus, and experiment 4, the compared mRNA expression between treatment and control groups using RT-qPCR.

### 2.13. Statistical Analysis

All values are presented as the mean ± standard error of mean (SEM), and *p* < 0.05 was considered statistically significant. In determining the optimal concentration, one-way analysis of variance and Tukey’s Multiple Comparison Test were used to compare the treatment and control groups. The independent sample *t*-test was used to compare the freeze-thaw quality parameters (motility, viability, membrane integrity, mucus penetration, and protamine deficiency) between treatment and control groups. Data were analyzed using SPSS 21.0 software (SPSS Inc., Chicago, IL, USA).

## 3. Results

### 3.1. Determination of Optimum Exosomal Protein Concentration

Sperm treated with 50 µg/mL exosomal protein exhibited higher motility (77.8 ± 0.2%) than untreated controls and those treated with the other concentrations (control, 75.1 ± 0.6%; 25 µg/mL, 73.8 ± 0.3%; 100 µg/mL, 73.9 ± 0.5%; Table 2). However, differences in the linearity, straightness, and ALH were non-significant between the groups. Data related to the other motion characteristics is included in Appendix A.

The percentage of live sperm was significantly higher in samples treated with 50 µg/mL exosomal proteins (68.6 ± 0.1%) than 25 µg/mL (64.8 ± 3.5%) or 100 µg/mL (64.9 ± 0.3%), (Figure 2). However, the difference between controls and treated samples (50 µg/mL) was not significant.

### 3.2. Effect of Exosomes on Post-Thaw Sperm Motility, Kinematic Parameters, and Viability

The post-thaw percentage of motile sperm was significantly enhanced in exosome-treated sperm than in control (56.8 ± 0.3% vs. 47.2 ± 0.3%, respectively) (Table 3). The percentage of linearity and straightness did not differ significantly between the treatment and control groups. However, ALH was significantly higher in treated sperm than in controls (3.2 ± 0.0 µm vs. 2.5 ± 0.0 µm, respectively). The percentage of live sperm statistically higher in treated than in untreated samples (55.9 ± 0.4% vs. 45.5 ± 0.4%, respectively) (Table 3). Data related with the motion characteristics is include in Appendix A.

### 3.3. Effect of Exosomes on the Integrity of the Plasma Membrane and Acrosome

HOS assay of post-thaw sperm showed that supplementation of buffer 2 with 50 µg/mL of exosomal proteins results in improved membranal integrity. The percentage of sperm with an intact plasma membrane was statistically higher in treated than in control samples (55.6 ± 0.5% vs. 47.8 ± 0.3%, respectively) (Table 3).

FITC-PNA staining of post-thaw semen showed that exosomes treatment results in significantly reduced sperm numbers with a damaged acrosome. The fraction of sperm with an intact acrosome was significantly higher in exosome-treated than control samples (60.4 ± 1.1% vs. 48.6 ± 0.4%, respectively) (Table 4).

### 3.4. Effect of Exosomes on Mucus Penetration

Mucus penetration test results reveal significantly higher sperm counts in treated than control samples at both the 1-cm and 3-cm marks (1 cm, 151.0 ± 0.6 vs. 136.5 ± 0.6; 3cm, 59.2 ± 0.3 vs. 47.7 ± 0.4, respectively) (Table 4). 

### 3.5. Effect of Exosomes on Chromatin Integrity and Gene Expression

Protamine-deficient sperm were identified by staining with CMA_3_. The post-thaw evaluation showed a significantly lower percentage of protamine-deficient in exosome-treated than control samples (24 ± 1.3% vs. 34.5 ± 0.7, respectively) (Figure 5). 

Post-thaw assessment of mRNA transcripts showed the significantly higher expression level of genes related to the repair of the plasma membrane (*ANX 1*, *DYSF*, *FN 1*) and chromatin components (*H3*, *HMGB*
*1*) in exosome-treated than control samples (Figure 6). However, the exosome-treated sperm showed a significant reduction in the expression of the mitochondrial ROS modulator (*ROMO 1*) gene. 

## 4. Discussion

The regenerative potential of MSCs is based on their ability to differentiate into various cell types and communicate with the surrounding cells through paracrine mechanisms. These paracrine mechanisms involve the production of different signaling factors in the form of membrane bounded vesicles [27,28]. These vesicles perform specialized functions including intercellular signaling and immune system regulation [17]. They exert their effect through the mediation of different protective and regenerative mechanisms at the cellular and molecular levels. Therefore, we attempted to evaluate the protective and regenerative potential of the exosomes isolated from conditioned media of canine Ad-MSCs in preserving the structural and functional integrity of canine sperm during cryopreservation.

Progressive motility and viability are considered key factors in the conservation of sperm function [29]. Thus, sperm fertility can be assessed through the estimation of motility, kinematic parameters, and viability. Supplementation with an optimal concentration of exosomes (50 µg/ mL) significantly increased the post-thaw motility, the ALH, and the fraction of live sperm. However, linearity and straightness percentages showed no significant difference between treated samples and controls (Table 3). Recent reports have demonstrated that treatment of cat sperm with exosomes isolated from oviductal fluids resulted in improved sperm motility [30]. In addition, freezing of rat sperm supplemented with MSCs derived microvesicles showed better post-thaw motility and viability [1]. An increase in the ALH along with a decrease in linearity or straightness indicates hyperactivation [31], which is reported to be significantly associated with the pregnancy rate [32]. Hyperactivation of exosomal protein-treated sperm was indicated by the results of the mucus penetration test. Significantly higher numbers of sperm were displaced to 1 cm and 3 cm in treated than in control sperm (Table 4). The mucus penetration test shows the capacity of sperm to penetrate surrogate cervical mucus and is commonly used to assess in-vitro and in-vivo fertilization and pregnancy rates [33].

Cellular architecture is highly dependent on the integrity of membranes. For sperm survival [34] and function [35], the integrity of the plasma and acrosomal membranes is critically important. These membranes are the primary sites that respond to environmental changes, undergoing severe damage during freezing [34]. The post-thaw evaluation showed that compared to untreated controls, a significantly higher percentage of exosomal-protein treated sperm had intact plasma and acrosome membranes. This protective effect of exosomal proteins toward the sperm plasma membrane is also reflected by the significantly higher motility and viability observed in treated sperm (Table 3). Ferraz et al. [30] also reported that sperm acrosome integrity was maintained following the incubation of cat sperms with exosomes. This increased membrane integrity may have resulted from the activity of exosomal signaling proteins that initiated membrane repair or protected sperm from oxidative damage. To investigate this possibility, we analyzed the gene expression of the membrane repair-related genes *ANX 1*, *DYSF*, and fibronectin *FN 1* and *ROMO 1* gene, which is responsible for generating mitochondrial ROS [36]. The expression levels *ANX 1*, *DYSF*, and *FN 1* were significantly higher in treated than in control sperm, while the expression of *ROMO 1* was significantly lower in treated sperm (Figure 6). In-vitro investigations of skeletal muscle suggest that an association between annexins and dysferlin in skeletal muscle repair [37]. Fibronectin regulates several cellular events and plays a vital role in the maintenance and repair of damaged tissues [38]. Different reproductive functions including activation of the proteasome, acrosomal reaction [39], capacitation [40], germ interaction [41], and embryonic development [42], are also mediated by fibronectin. 

Sperm chromatin is also negatively influenced by freezing ultimately resulting in reduced fertility, low-quality embryos [43], and several genetic disorders [44,45,46]. CMA_3_ staining of post-thaw sperm revealed a significantly higher percentage of protamine-deficient sperm in control than in treated sperm (Figure 5). This increase in nuclear integrity likely results from the elevated expression of proteins actively involved in the repair of nuclear material, including *H3* and *HMGB 1* (Figure 6). Recently, H3 [47] and HMGB [48] proteins have been isolated and identified in the exosomes. In mammals, sperm DNA is wrapped around the core histone H3, [49]. Variants of histone H3 are reported to be actively involved in the repair of damaged DNA. HMGB 1 is reported to be involved in a wide range of cellular processes, including the regulation of chromatin structure and transcription. HMGB 1 can bind to the damaged DNA, stabilizing it and increasing the mobility of chromatin. In addition, HMGB 1 is involved in the nucleotide excision repair [50].

## 5. Conclusions

Treatment with exosomal proteins maintains the integrity of the plasma membrane as well as the chromatin material of canine sperm during cryopreservation. This protective and regenerative potential of exosomes was further demonstrated by increases in standard indicators of sperm quality including progressive motility, ALH, viability, the integrity of membranes and the nucleus. The exosomal signaling factors derived from Ad-MSCs including both mRNA and protein are proposed to be responsible for this improvement. This study identifies several signaling factors that may contribute to sperm quality. However, further investigations are required to identify and evaluate the effects of a variety of unidentified signaling factors present in exosomes that may increase the quality of post-thaw semen.

## Figures and Tables

**Figure 1 animals-09-00865-f001:**
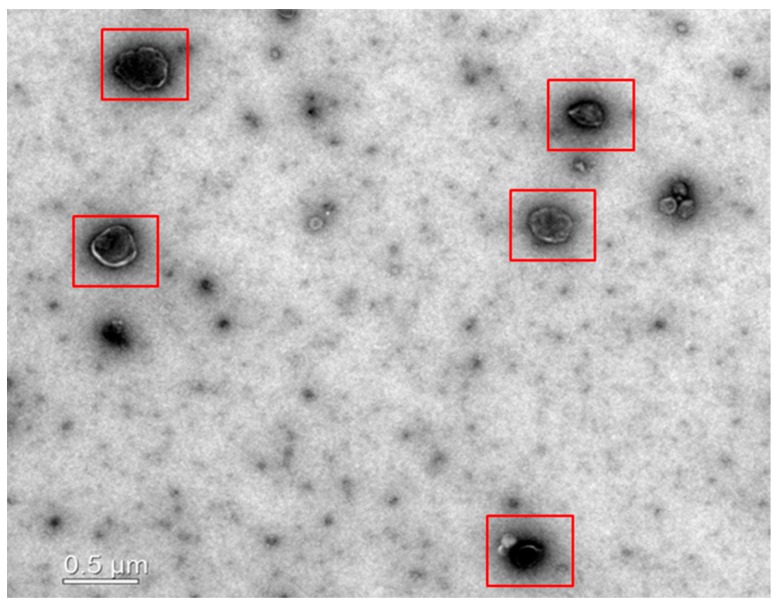
Transmission electron micrograph showing exosomes (red boxes) isolated from conditioned medium derived from canine adipose-derived mesenchymal stem cells (Ad-MSCs).

**Figure 2 animals-09-00865-f002:**
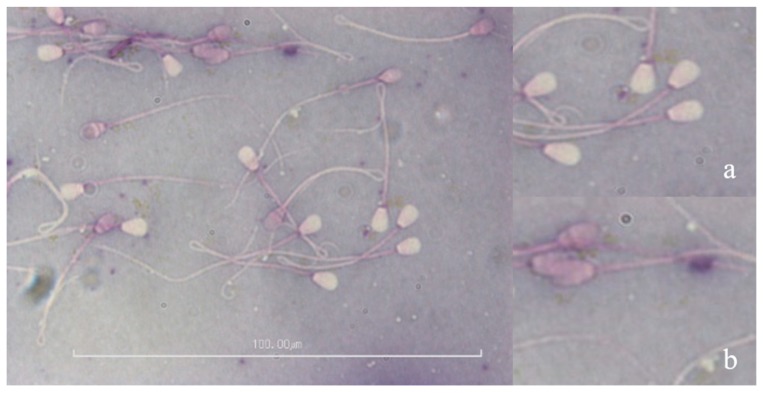
Image of eosin-nigrosin stained sperm showing live and dead sperm. Insets: magnified view of (**a**) live sperm, and (**b**) dead sperm.

**Figure 3 animals-09-00865-f003:**
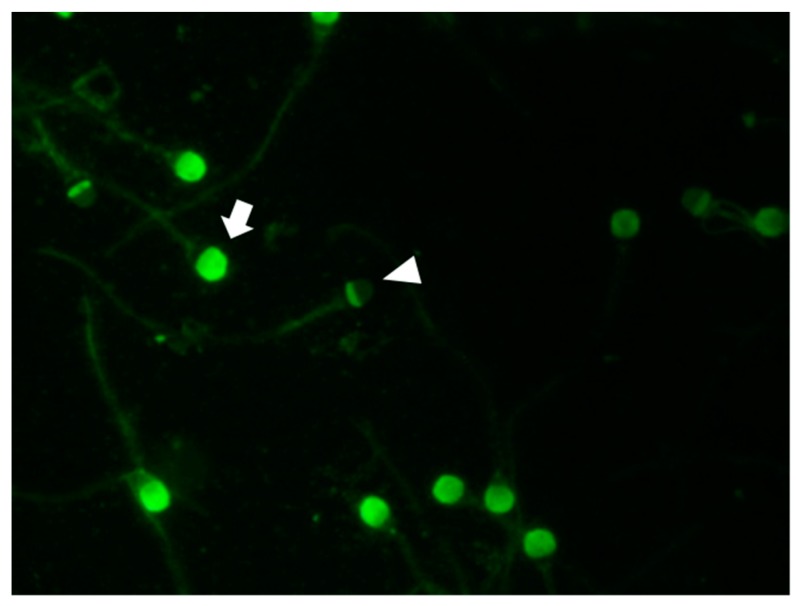
Sperm stained with fluorescein isothiocyanate peanut agglutinin to show sperm with intact (arrow) and damaged (arrowhead) acrosomes.

**Figure 4 animals-09-00865-f004:**
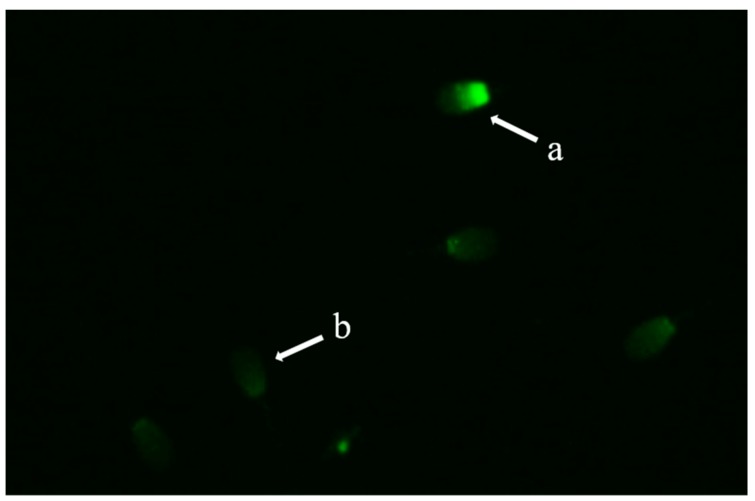
Chromomycin A_3_ staining of sperm chromatin to indicate protamine-deficient (a) and normal sperm (b).

**Figure 5 animals-09-00865-f005:**
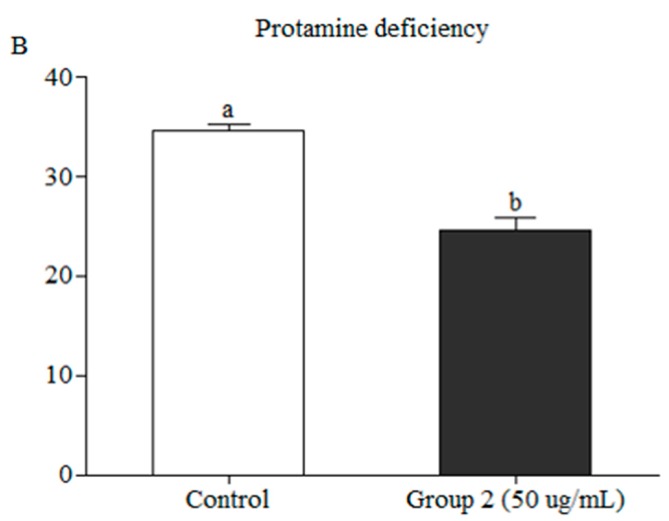
Quantification of Chromomycin A3 in exosome treated and untreated sperm (*p* < 0.05).

**Figure 6 animals-09-00865-f006:**
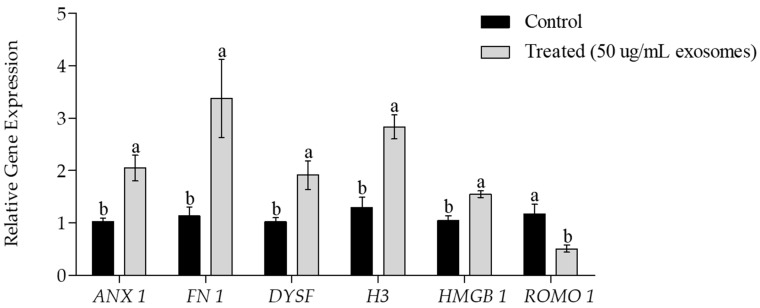
Expression of the plasma membrane repair-related genes annexin 1 (*ANX 1*), fibronectin (*FN 1*), and dysferlin (*DYSF*); the chromatin repair-related genes histone H3 (*H3*) and high mobility group protein B (*HMGB*); and a mitochondrial ROS modulator (*ROMO 1*) as determined by the real-time RT-qPCR in exosomes treated and untreated sperm. Data are presented as the mean ± SEM. Different lower case letters, a, or b represent a significant difference *p* < 0.05.

**Table 1 animals-09-00865-t001:** Primer sequences used for gene expression analysis.

Gene	Primer Sequence (5′-3′)	Product Size (bp)	NCBI Accession No.
*BACT*	F: GAGGCATCCTGACTCTGA	87	XM_544346.3
	R: TCTGGCACCACACTTTCT		
*ANX1*	F: GAAGCTCTGAAGAAAGCCC	128	NM_001286970.1
	R: GTGTCTTCATCAGTTCCAAGG		
*DYSF*	F: TGGATCAGAGTGGCGTCC	127	XM_003432223.4
	R: GACAGCAGCTTTCTGGCT		
*FN1*	F: ATAGCTGGCTGTTACGAC	74	XM_022415242.1
	R: GCATTTCCCAGGTAGGTG		
*H3*	F: CGGTGACTGACACGCGAC	136	XM_022404950.1
	R: GGTTCAAGGCCTGCTCCAAC		
*HMGB*	F: ATATTGCTGCGTACCGAG	64	XM_022409535.1
	R: TCAGCCTTGACAACTCCC		
*ROMO1*	F: CTACGTGCTCCCGGAAGT	100	XM_534406.6
	R: TCGCTCAGTTCTACGTCTCAC		

F, forward; R, reverse. Annexin 1, *ANX1*; histone H3, *H3*; fibronectin 1, *FN1;* high mobility group protein B, *HMGB*; dysferlin, *DYSF*; mitochondrial ROS modulator, (*ROMO1*).

**Table 2 animals-09-00865-t002:** Determination of optimal concentration of exosomal proteins for cryopreservation of dog sperm after incubation 4 °C for 8–10 h.

Groups	Motility (%)	Linearity (%)	Straightness (%)	ALH (µm)	Live sperm (%)
Control	75.1 ± 0.6 ^b^	26.4 ± 1.0	47.1 ± 0.7	4.4 ± 0.3	67.9 ± 0.3 ^a^
25 µg/ mL	73.8 ± 0.3 ^b^	24.5 ± 0.6	45.9 ± 0.7	4.7 ± 0.1	64.8 ± 3.5 ^b^
50 µg/ mL	77.8 ± 0.2 ^a^	26.3 ± 0.3	47.5 ± 1.1	5.0 ± 0.1	68.6 ± 0.1 ^a^
100 µg/ mL	73.9 ± 0.5 ^b^	26.6 ± 0.9	46.2 ± 1.6	4.7 ± 0.1	64.9 ± 0.3 ^b^

ALH, amplitude of lateral head displacement. Values with different superscripts letters in a column differ significantly (*p <* 0.05, *n* = 4).

**Table 3 animals-09-00865-t003:** Effects of exosomal protein treatment on the post-thaw motility, kinematic parameters, viability, and membrane integrity of dog semen.

Group	Motility (%)	Linearity (%)	Straightness (%)	ALH (µm)	Live Sperm (%)	Membrane Integrity (%)
Control	47.2 ± 0.3 ^b^	26.8 ± 1.0	51.6 ± 0.5	2.5 ± 0.0 ^b^	45.4 ± 0.4 ^b^	47.8 ± 0.3 ^b^
Treatment (50 µg/mL)	56.8 ± 0.3 ^a^	28.8 ± 1.3	50.5 ± 0.6	3.2 ± 0.0 ^a^	55.9 ± 0.4 ^a^	55.6 ± 0.5 ^a^

ALH, amplitude of lateral head displacement. Values with different superscripts letters in a column differ significantly (*p*
*<* 0.05, *n* = 4).

**Table 4 animals-09-00865-t004:** Effects of exosomal protein treatment on acrosome integrity and mucus penetration ability of post-thaw dog semen.

Groups	Integrity of Acrosome (%)	Number of Sperm Penetrating Mucus	
		1 cm	3 cm
Control	48.6 ± 0.4 ^b^	136.5 ± 0.6 ^b^	47.7 ± 0.4 ^b^
Treatment (50 µg/mL)	60.4 ± 1.1 ^a^	151.0 ± 0.6 ^a^	59.2 ± 0.3 ^a^

ALH, amplitude of lateral head displacement. Values with different superscripts letters in a column differ significantly (*p <* 0.05, *n* = 4).

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
