# Peer review of "Improved Post-Thaw Quality of Canine Semen after Treatment with Exosomes from Conditioned Medium of Adipose-Derived Mesenchymal Stem Cells"

_animals, 2019, doi:10.3390/ani9110865_

Round 1

Reviewer 1 Report

In this study, Qamar et al. investigated the effects of MSCs-derived exosomes on some parameters of frozen-thawed canine semen.

This is an interesting and promising study. It is worth noting that this study is one of the first trials to use the exosomes derived from MSCs to improve the cryotolerance of mammalian sperms (canine for this model).

The approach is novel and the methods are enough however, there are some flaws require revision.

Summary: Delete lines 13-16 to make it concise.

L16: Delete  etc. and modify the sentence.

L37-38: There is no data about ROS levels.

L66: from line 66 and throughout the manuscript, please replace EVs with Exosomes, where is applicable.

L95: replace “Petri” with “tissue culture” dish.

L99: was it 28 or24h or less, cells without serum for this long duration will cause cell death. If so, please insert a suitable citation to support this.

L107: use “exosomes pellets”. Were the pellets used immediately or kept frozen until usage?

L110: Correct "AD-MSCs”

L112: delete "using performed"

L128: add " or with.."

L131: please check the time, 4h is a short duration.

L133: correct "Sperms were"

L196: Replace the subheading “Relative quantitative PCR analysis”, L197: add “relative” before quantitative.

L201: delete “RNA samples” and replace with "total RNA was extracted..."

L204: Please check the name of the cDNA synthesis kit. Is it correctly typed?

L318-319: delete this sentence.

L147: correct “Minitube”

L152: add "previously." and remove the remained parts of the sentence.

L156: specify the HOS solution.

Figure 2: a, and b, do not match with the figure on the left side. So, please edit the figure labels to specifically show the determined image.

Figure 3: remove "a" and "b", the arrow and arrowhead are indicative.

Figure 4B: it should follow the text of the corresponding result (3.5.) not in the M&M.

In Table 1:

Where are sequences for the internal control gene (beta actin; ACTB)?

The given accessions are for the whole chromosomes, so follow what I found and correct the mistyped ones as follows:

ANX1>>NM_001286970.1

DYSF>>XM_003432223.4

FN1>>XM_022415242.1

HMBG>>XM_022409535.1

ROMO1>>XM_534406.6

And for “H3” Please check the sequences of F and R primers and the accession No., it is wrongly indicated.

In Table 2: Please specify the temperature and duration in the Table caption.

Do you think the range differences of 3-4% that was found in the studied parameters are convincing to depend on the chosen concentration (50ug/mL). I suggest reducing the tone of exaggeration of the results.

Table 3: Move Table 3 after the result 3.4.

Discussion:

Discussion requires to be more focused on the effects of exosomes and attribute the effects to the contents of the exosomes. The current discussion lacks supportive references about the possible effects of exosomes on sperm parameters.

I suggest this recent reference to support the discussion:

Ferraz et al. 2019 Oviductal extracellular vesicles interact with the spermatozoon’s head and mid-piece and improves its motility and fertilizing ability in the domestic cat. Scientific Reports volume 9, Article number: 9484

Author Response

Comment 1: This is an interesting and promising study. It is worth noting that this study is one of the first trials to use the exosomes derived from MSCs to improve the cryotolerance of mammalian sperms (canine for this model). The approach is novel and the methods are enough however, there are some flaws require revision.

Response: Thank you for your assessment and useful suggestions for improving our manuscript. 

Comment 2: Delete lines 13-16 to make it concise.

Response: We agree to the reviewer's advice and deleted the lines 13-16 to make the summary more concise.

Comment 2: L16: Delete etc. and modify the sentence.

Response: We agreed with the advice and adopted the changes.

Comment 3: L37-38: There is no data about ROS levels.

Response: Thank you for pointing out this error. We agree with the observation of the respected reviewer as there was no assessment of the ROS level. But we used this statement on the basis of low expression of the ROMO 1 gene, an indication of lower ROS production. 

Comment 4: L66: from line 66 and throughout the manuscript, please replace EVs with Exosomes, where is applicable.

Response: We appreciated and followed the suggestion of the respected reviewer by replacing EVs with exosomes where it was applicable.

Comment 5: L95: replace “Petri” with “tissue culture” dish.

Response: Thanks for your advice. We have adopted the recommended change as "tissue culture dish" is a more precise and clear word than "petri dish".

Comment 6: L99: was it 28 or24h or less, cells without serum for this long duration will cause cell death. If so, please insert a suitable citation to support this.

Response: We appreciate your comment about the starvation period. We have supported our procedure of using 48 h starvation by additng a citation in the context as per the recommendation of the reviewer.

Comment 7: L107: use “exosomes pellets”. Were the pellets used immediately or kept frozen until usage?

Response: Respected reviewer we agree with your suggestion of "exosomes pellet". We have also mentioned that the pellet was immediately used as it was extracted from the conditioned medium.

Comment 8: L110: Correct "AD-MSCs”.

Response: We are thankful to the reviewer for pointing out our mistake in order to keep uniformity in the manuscript. We have made the correction.

Comment 9: L112: delete "using performed"

Response: Thank you for pointing out our mistake. We have deleted "using performed".

Comment 10: L128: add " or with.."

Response: We have made the correction according to the respected reviewer's advice.

Comment 11: L131: please check the time, 4h is a short duration.

Response: We agree with the reviewer's comment about this mistake. We have made the correction.

Comment 12:   L133: correct "Sperms were"

Response: We agreed to the recommendation of the reviewer and made changes accordingly.

Comment 13: L196: Replace the subheading “Relative quantitative PCR analysis”, L197: add “relative” before quantitative.

Response: We agreed to the recommendation of the respected reviewer and adopted the changes. 

Comment 14: L201: delete “RNA samples” and replace them with "total RNA was extracted..."

Response: We appreciated and adopted the recommendations of the respected reviewer.

Comment 15: L204: Please check the name of the cDNA synthesis kit. Is it correctly typed?

Response: Following the advice of the reviewer, we have checked and confirmed the name of the cDNA synthesis kit (Compact cDNA synthesis kit). 

Comment 16: L318-319: delete this sentence.

Response: Sentence was deleted as recommended by the respected reviewer.

Comment 17: L147: correct “Minitube”

Response: We appreciated and adopted the recommendations of the respected reviewer.

Comment 18: L152: add "previously." and remove the remained parts of the sentence.

Response: We agree to the advice of the reviewer and adopted the recommendations of the respected reviewer.

Comment 19: L156: specify the HOS solution.

Response: We agree with this suggestion of the reviewer and incorporated the specification of the HOS solution in the context.

Comment 20: Figure 2: a, and b, do not match with the figure on the left side. So, please edit the figure labels to specifically show the determined image.

Response: We agree to the comments of the reviewer about Figure 2. We have edited the figure as per recommendations. 

Comment 21: Figure 3: remove "a" and "b", the arrow and arrowhead are indicative.

Response:  We agreed to the reviewer's advice and changed the Figure 3 accordingly.

Comment 22: Figure 4B: it should follow the text of the corresponding result (3.5.) not in the M&M. 

Response: We incorporated the change following the reviewer's advice.

Comment 23: In Table 1: Where are sequences for the internal control gene (beta-actin; ACTB)? The given accessions are for the whole chromosomes, so follow what I found and correct the mistyped ones as follows:

ANX1>>NM_001286970.1

DYSF>>XM_003432223.4

FN1>>XM_022415242.1

HMBG>>XM_022409535.1

ROMO1>>XM_534406.6

And for “H3” Please check the sequences of F and R primers and the accession No., it is wrongly indicated.

Response: We are thankful to the reviewer for a detailed evaluation of our manuscript,  pointing out our mistakes and suggesting some additions. We have made the corrections and added the sequence of beta-actin in the manuscript.

Comment 24: In Table 2: Please specify the temperature and duration in the Table caption. Do you think the range differences of 3-4% that were found in the studied parameters are convincing to depend on the chosen concentration (50ug/mL)? I suggest reducing the tone of exaggeration of the results.

Response: Following the advice of the reviewer, we have made the adjustments.

Comment 25: Table 3: Move Table 3 after the result 3.4.

Response: We agreed to the suggestion made by the reviewer and moved the table after 3.4.

Comment 26: Discussion requires to be more focused on the effects of exosomes and attribute the effects to the contents of the exosomes. The current discussion lacks supportive references about the possible effects of exosomes on sperm parameters.

Response: We agree to the reviewer's advice, as the addition of references related to exosomal effects on sperm will make the discussion better than before. So we have cited a few references in this regard.

Reviewer 2 Report

The manuscript is very interesting. However there are several important points in material and methods to correct:

How many ejaculates were obtained/male? Is very important that you explain this for each experiment and that this was reflected in the results and tables (n). Line 238 (results) "the effect of exosomal proteins... by analyzing four independent replicates". This is material and methods. And 4 replicates means 1 ejaculate for each male? Explain this better. Lines 134-135. Explain the motility descriptors analyzed by your CASA system. Only motility, linearity, straightness and ALH? And why not velocities, BCF (beat cross frequency), wobble (WOB) or Progressive motility? You are explaining in line 292 (discussion) the importance of Progressive motility. Statistical analysis. Did you use a normality test before other statistical analysis?

Other comments:

. Results. Why tables 3a and 3b?. It will be better 3 and 4. On the other hand, the table legend is 3 in both cases!

. Discussion. Line 297. ALH increases in hyperactivated spermatozoa but not only in this case.

Author Response

Comment 1: The manuscript is very interesting. However, there are several important points in material and methods to correct.

Response: We really appreciate both the positive comments as well as corrections/suggestions from the respected reviewer for the improvement of the article.

Comment 2: How many ejaculates were obtained/male? Is very important that you explain this for each experiment and that this was reflected in the results and tables (n).

Response: We agree to the reviewer's advice regarding mentioning the number of ejaculates/male. In each replicate, we used 1 ejaculate/male and the total number of replicates was 4. We have included this information in tables and results.

Comment 3: Line 238 (results) "the effect of exosomal proteins... by analyzing four independent replicates". This is material and methods. And 4 replicates means 1 ejaculate for each male? Explain this better.

Response: We admit that this statement is not clear and should be a part of the material and methods. We have moved it from results and tried to make it more clear. Actually, we have mentioned in the material and methods that after semen collection from each male semen was pooled together before further processing to eliminate individual variations. So 1 replicate does not mean ejaculate from a single male.

Comment 4:   Lines 134-135. Explain the motility descriptors analyzed by your CASA system. Only motility, linearity, straightness and ALH? And why not velocities, BCF (beat cross frequency), wobble (WOB) or Progressive motility? You are explaining inline 292 (discussion) the importance of Progressive motility.

Response: Thanks for pointing out the deficiencies. We have provided the required information in the form of supplementary data tables. In the discussion, we tried to focus on possible reasons for the improvement of sperm quality but after the advice of the respected reviewer, we have provided the data.

Comment 5: Statistical analysis. Did you use a normality test before other statistical analysis?

Response: Thanks for inquiring about statistical analysis. We have used the normality test.

Comment 6: Results. Why tables 3a and 3b? It will be better 3 and 4. On the other hand, the table legend is 3 in both cases!

Response: We agree with the mistake identified by the reviewer for a better understanding of the manuscript. We changed the number as well as the table legends to eliminate any chance of confusion. 

Comment 7: Discussion. Line 297. ALH increases in hyperactivated spermatozoa but not only in this case.

Response: We agree with the reviewer's comment, that ALH alone is not an indication of hyperactivation.  An increase in ALH should be accompanied by a decrease in linearity and straightness. In our data, there was a significant increase in ALH but Linearity and straightness remained nonsignificant. 
